# Labor market segmentation and the gender wage gap: Evidence from China

**Mingming Li** [1,2]*, **Yuan Tang**[3], **Keyan Jin**[4]

1 Department of Sociology, Faculty of Social and Political Sciences, University of Innsbruck, Innsbruck, Austria, 2 Institute for Common Prosperity and Development, Zhejiang University, Hangzhou, China, 3 The Marxist College, Zhejiang University, Hangzhou, China, 4 Department of Quantitative Methods in Economics and Business, University of Granada, Granada, Spain

* li_mingming@phd.ceu.edu

## Abstract

Although the Chinese government has implemented a variety of measures, the gender wage gap in 21st century China has not decreased. A significant body of literature has studied this phenomenon using sector segmentation theory, but these studies have overlooked the importance of the collective economy beyond the public and private sectors. Moreover, they have lacked assessment of the gender wage gap across different wage groups, hindering an accurate estimation of the gender wage gap in China, and the formulation of appropriate recommendations. Utilizing micro-level data from 2004, 2008, and 2013, this paper examines trends in the gender wage gap within the public sector, private sector, and collective economy. Employing a selection bias correction based on the multinomial logit model, this study finds that the gender wage gap is smallest and most stable within the public sector. Furthermore, the private sector surpasses the collective economy in this period, becoming the sector with the largest gender wage gap. Meanwhile, a recentered influence function regression reveals a substantial gender wage gap among the low-wage population in all three sectors, as well as among the high-wage population in the private sector. Additionally, employing Brown wage decomposition, this study concludes that inter-sector, rather than intra-sector, differences account for the largest share of the gender wage gap, with gender discrimination in certain sectors identified as the primary cause. Finally, this paper provides policy recommendations aimed at addressing the gender wage gap among low-wage groups and within the private sector.

## 1 Introduction

In the past few decades, the Chinese government has been committed to promoting gender equality and reducing the gender wage gap. This includes the formulation and implementation of laws and regulations against gender discrimination, encouraging women to pursue higher education and vocational training, providing more employment and promotion opportunities, urging employers to offer equal wages and opportunities, and ensuring that women receive fair treatment in the workplace [1–4]. However, these attempts have not worked as expected.

**Data Availability Statement:** All relevant data are within the manuscript and its Supporting Information files.

**Funding:** Sources of funding1: The project is supported by Publishing fund of the University of Innsbruck and Department of Sociology in

University of Innsbruck Receiver: Mingming Li Sources of funding2:China Scholarship Council (CSC), grant number: 202106980007 Receiver: Keyan Jin. CSC supports Keyan Jin living costs.

**Competing interests:** The authors have declared that no competing interests exist.

According to the World Economic Forum "Global Gender Gap Report 2020", the global ranking of China in terms of gender pay equality has dropped from 57th in 2006 to 106th in 2020. This shows that the gender wage gap in China is increasing [5].

Besides factors like industry [6–8], urban and rural areas [9–11], party membership [12], and occupation [13, 14], in recent years the focus of discussions regarding the widening gender wage gap has gradually shifted toward sector segmentation theory and related empirical studies [11–14]. Sectoral segmentation theory is an explanatory framework for understanding and explaining the existence of gender wage gaps in different industries or sectors [15]. The theory suggests that the characteristics and nature of industries or sectors may lead to the existence of a gender wage gap, and this gap is related to the allocation and positioning of gender in the labor market. In China, different sectors reflect significant differences between genders in terms of occupational choices, job hierarchies, working conditions and benefits, from both inter-sector and intra-sector perspectives [16–18]. However, there are currently no systematic and clear explanations of how sector segmentation affects the gender wage gap, due to theory limitations, sample availability, research methods, and often-changing labor policies. Therefore, this paper tries to understand the role of sector segmentation in the gender wage gap and its change trend in the context of China, addressing the limitations of current research.

Firstly, the bulk of the existing literature focuses on the gender wage gap caused by dividing the labor market between the public and private sectors. Although this topic has been discussed for a long time, clear conclusions have not yet been reached. The public sector, which is usually regulated and governed by the government, tends to receive greater attention regarding the gender wage gap. Government departments usually emphasize the principles of fairness and equality in the recruitment and promotion process, and strive to reduce gender discrimination [19]. The gender wage gap in the private sector is more influenced by market mechanisms. Private companies usually pay more attention to economic efficiency and profit maximization and may have a lower level of concern for the gender wage gap. In market competition, gender discrimination and professional bias may cause women to face more obstacles in terms of promotion and securing high-paying positions, thereby increasing the gender wage gap. However, some scholars argue that a gender wage gap still exists in the public sector, although it is smaller. This may be due to the fact that men still hold the majority of senior positions in the public sector, while women tend to be more concentrated in lower-level positions [4]. At the same time, the gender wage gap in the private sector is not entirely caused by gender discrimination. Based on the theory of human capital, it is affected by various factors such as education, work experience, and job choices. Some researchers suggest that the high concentration of women in low-paid private sector industries is more due to their own choices than gender discrimination [20].

Secondly, previous research has completely ignored the role played by the collective economy in the gender wage gap. Besides the public and private sectors, the collective economy has long been an indispensable component of the Chinese economy [11, 21, 22]. The collective economy in China is a form of economic organization in which farmers in rural areas jointly manage farmland, forest land and other resources through collective ownership [23]. Although it no longer accounts for a large proportion of the Chinese economy, it still plays an important role in the modernization of agriculture and the growth of agricultural wages [24]. It is also highly relevant to village autonomy and collective prosperity. An introduction to the collective economy can be found in S1 Text. Since the collective economy is the traditional employment structure in rural areas, gender role stereotypes persist, making women more susceptible to gender discrimination in the rural collective economy [14, 25]. For example, in rural cooperatives, men tend to occupy a larger proportion of decision-making and high-paying positions, while women are more commonly engaged in low-paying and non-leadership positions. The

primary objective of the collective economy is to meet the economic interests of farmers and to further rural economic development; consequently, salary levels are relatively low. This may lead to a lower salary level for women in the collective economy, thereby increasing the gender wage gap. In addition, due to the relatively weak welfare security system in rural areas, women may face unequal treatment in terms of wages, social insurance, and medical coverage. In rural areas, career development opportunities in the collective economy are relatively limited, and promotion channels are narrow. This may restrict the career development of women in the collective economy, resulting in a widening of the gender wage gap. Men are more likely to enter management and leadership positions in the rural collective economy, while women are more engaged in grassroots and supportive work [26].

Thirdly, the majority of empirical studies only consider average wage differences. There is a lack of sufficient understanding of the discrepancies in the wage gap between different sectors and across various wage distributions. Melly [27] proposes that using regression and traditional methods may yield different outcomes. By specifically examining the gender wage gap across various quantiles, a more nuanced understanding of inequality within the wage distribution can be achieved. Understanding these differences can aid in formulating more targeted policies and measures to diminish the gender wage gap, as well as highlighting structural issues [28]. For example, if there is a significant gender wage gap at lower wage levels, this may imply that women are more likely to be affected by the wage gap in low-paying industries or positions with poorer working conditions. This understanding helps to identify potential inequalities and discrimination issues and allows for appropriate measures to be taken [29].

Fourthly, the gender wage gap caused by sectoral segmentation in the 21st century has changed with the passage of time. However, the relevant literature lacks a trend analysis of the gender wage gap in China over time. In the early 2000s, the gender wage gap in China was generally large, and was further exacerbated by sector segmentation. The public sector had a smaller gender wage gap compared to the private sector and the collective economy. This was mainly influenced by traditional views and social structures, whereby women encountered limitations and discrimination in certain industries and positions. In the mid-to-late 2000s, the Chinese government began to focus more on gender equality issues and adopted a series of policies aimed at reducing the gender wage gap [4]. The government fortified legal frameworks against gender discrimination, elevated gender equality education and awareness, and advocated for women's participation across all sectors and fields [30]. The government in the 2010s further increased its policy support for gender equality, provided equal employment opportunities and promotion mechanisms, and improved the treatment of women in the working environment [31]. This led to a gradual decrease in the gender wage gap in the public sector. Therefore, it is necessary to study the gender wage gap caused by sector segmentation in different time periods.

Additionally, the gender wage gap in China is influenced by both intra-sector and inter-sector differences [32]. Wage gaps between genders may manifest in the public sector, private sector, and collective economy due to a range of factors such as industry-specific traits, occupational division of labor, and gender discrimination [33]. For example, in some traditional industries, women face restrictions in terms of job promotion and salary due to the influence of social concepts and the division of traditional roles. Gender discrimination may also exist in certain industries in the private sector, resulting in women receiving lower salaries than men. The gender wage gap is wider in some sectors compared with others. In China, the public sector typically places more emphasis on gender equality and pay equity compared to the private sector and the collective economy. The government has adopted a series of policy measures in the public sector to provide equal employment opportunities, promotion

mechanisms, and welfare benefits, in order to reduce the gender pay gap. Therefore, it is necessary to explore the gender wage gap from both inter-sector and intra-sector perspectives.

This paper studies both the level of gender inequality brought about by sector segmentation in the Chinese labor market, and its development trends. The main data used in this study are cross-sectional data from the 2004, 2008, and 2013 Urban Household Survey (UHS) and the Labor Statistical Yearbook.

Specifically, this paper addresses the following core questions:

1. What are the wage gaps between males and females in the public sector, private sector, and collective economy?

2. What are the wage gaps between males and females across different wage groups within these sectors?

3. How have these gender wage gaps evolved over time?

4. What roles do intra-sector and inter-sector differences play in causing gender wage gaps?

This article finds that a gender wage gap exists in all three sectors from 2004 to 2013. The gap varies across different quantiles and undergoes changes over time. Specifically, the gender wage gap in the public sector is consistently the smallest and most stable of the three sectors. However, during this period, the private sector surpasses the collective economy and becomes the sector with the largest gender wage gap. At the same time, the gender wage gap within low-wage groups in all three sectors is significant. In addition, the gender wage gap among high-wage employees in the private sector is also large. Finally, this study finds that differences between sectors (inter-sector) rather than within sectors (intra-sector) are the main cause of the gender wage gap, and this is mainly because of unexplained discrimination.

This study contributes to the existing literature on the gender wage gap in China in several ways. Firstly, this article validates the existence of sector segmentation in China's labor market and establishes that the theory of human capital continues to be applicable. Secondly, the article confirms that the gender wage gap in the private sector has overtaken that in the collective economy, designating the private sector as the domain with the widest wage gap and thereby serving as a crucial focus for future policy efforts. Thirdly, the study reveals that the collective economy still holds significant sway, and the gender wage gap within this sector also influences the overall gap. Fourthly, there exists a large gender wage gap within low-wage groups in all three sectors; hence, future policies should lean more towards protecting the rights and welfare of low-wage women.

The remaining structure of this paper is as follows: Section 2 outlines the developmental background and wage-determination mechanisms of different sectors in China; Section 3 introduces the representative theories and relevant empirical literature; Section 4 presents the data and theoretical framework for empirical analysis, along with descriptive statistics; Section 5 offers the results of regression analyses and decomposition of wage gaps, followed by discussion; Section 6 concludes the paper, providing corresponding policy recommendations, and identifying limitations and directions for future research.

## 2 Sector segmentation in China

Unlike developed countries, the labor market in China is composed of three sectors: the public sector, the private sector, and the collective economy. Differences in segmentation occur as a cumulative result of various factors. In terms of operational objectives, the public sector assumes important functions and responsibilities in the Chinese economy, including the provision of public services, social security, education, healthcare, and infrastructure construction

[34–36]. The private sector consists of businesses and organizations that operate in accordance with the principles of a market economy, whose purpose is to pursue profits and economic growth [37, 38]. The collective economy is a special sector that mainly involves rural collective economic organizations and cooperatives. It is closely related to the rural land system and farmers' organizations.

From a historical and cultural standpoint, heritage has notably influenced the development and shaping of different sectors. Traditional Chinese culture emphasizes the value of public interests and collectivism, with the government playing a pivotal role in social and economic spheres. Following the era of reform and opening up, the public sector has been instrumental in regulating the Chinese economy, providing basic public services, and maintaining societal stability. The private sector has rapidly emerged in a market-oriented economic environment, contributing to economic growth, job creation, and providing diverse products and services. The collective economy plays a role in supporting farmers' livelihoods and promoting rural development [39, 40]. It diverges from the public and private sectors in aspects such as ownership, operational goals, organizational structures, managerial mechanisms, labor force requirements, and funding sources. The collective economy primarily aims to fulfill the economic interests of farmers and promote rural economic development. The public sector is dedicated to providing public services and meeting basic social needs, while the private sector pursues economic profit and commercial success.

In China, wage determination mechanisms differ between the public sector, private sector, and collective economy, contributing to the gender wage gap. In the public sector and the collective economy, wage policies are usually formulated by the government, aimed at achieving fairness and equality in remuneration. On the contrary, wage determination in the private sector is largely influenced by market forces and prioritizes profitability and competitiveness. If gender discrimination or biases are present in these sectors, wage setting could favor a specific gender, thereby widening the gender wage gap [41]. Secondly, the criteria for job evaluations and promotions in various sectors also impact the gender wage gap. Within the public sector, clear and objective criteria often exist, which help to mitigate the influence of subjective factors on the gender wage gap. However, in the private and collective economic sectors, promotions may largely depend on individual performance and internal relationships, potentially leading to gender discrimination and consequently, increases in the gender wage gap [42]. Additionally, the provision of welfare and social security measures by different sectors also influences the gender wage gap. The public sector usually offers equal benefits like health insurance, pensions, and maternity leave. These measures mitigate the economic losses that women may incur due to family responsibilities throughout their careers [43]. However, the private sector and the collective economic sector may provide fewer welfare benefits, which could contribute to an increase in the gender wage gap. Additionally, the level of compliance with gender equality policies and regulations in various sectors also has an impact on the gender wage gap. The public sector is typically regulated and controlled by the government, making it more likely to adhere to gender equality policies and regulations. On the other hand, the private and collective economic sectors may have shortcomings in implementing these policies and regulations, thereby increasing the gender wage gap [44].

## 3 Literature review

### 3.1 Wage gap theory

**3.1.1 Theories of labor market segmentation.** In human capital theory, the role of education investment is pivotal. With the emergence of contemporary human capital theory, governments worldwide have increasingly focused on education, investing in human capital to

stimulate economic growth. In so doing, they have addressed numerous social challenges. However, gender-based wage gaps remain largely unchanged, as they are also influenced by the divergent distribution of employment between men and women across various sectors, industries, and occupations. Thus, human capital theory does not offer a comprehensive solution to this issue.

Devine [45] contends that the neoclassical principles governing labor markets have limitations in explaining the gender wage gap. Written in 1971, the book Internal Labor Markets and Manpower Analysis by Doeringer and Piore [46] serves as a seminal study in labor market segmentation theory. Their research on the labor market in Boston reveals that human capital theory falls short in explaining the gaps between high and low earners. Labor market segmentation is essential, subdividing the market into primary and secondary segments based on labor ability. Individuals enjoying favorable working conditions, high salaries, and ample promotion opportunities predominantly occupy the primary market, while those with lower socioeconomic status largely find themselves in the secondary market. Thus, labor market segmentation mirrors the economic and social statuses of workers.

Sectoral segmentation is a crucial element of labor market segmentation [47–50]. The public and private sectors can be used as analogies for the primary and secondary markets. For instance, the equilibrium wage in the private sector is determined primarily by the market, while the adjustment of wages in the public sector is influenced mainly by the government [49, 51, 52]. That is, whereas the public sector is arguably protected more by its egalitarianism, workers in the private sector are in a more competitive labor market. Sectoral segmentation brings forth different mechanisms of wage determination; therefore, it can distort the employment choices and wage distribution of male and female workers, which contributes to the gender wage gap [17, 53, 54].

**3.1.2 Human capital theory.** The concept of human capital has a long history. Although not explicitly named, Adam Smith writes in 1776 about the "acquired and useful abilities of all the inhabitants or members of society" [55]. Fisher [56] introduces the modern concept of human capital, further refined as a theory on the "economic value of education" by Schultz [57]. Mincer [58] notes that both schooling and work experience directly impact individual earnings and develops a function based on human capital theory to depict the correlation between earnings, educational attainment, and work experience. Becker [59] expands the scope to include not only formal education but also on-the-job training and labor mobility. Becker reasons that both male and female workers freely allocate their labor time in line with market principles, which accounts for the uneven occupational distribution by gender and the resultant wage gap.

**3.1.3 Compensating wage differentials theory.** This theory posits that variations in job nature directly affect labor compensation [60]. Even with identical skills and abilities, workers may receive different wages due to disparate working conditions. For example, those employed in less favorable conditions should command higher wages to compensate for these drawbacks. Compensatory wages serve to motivate workers to accept challenging or hazardous positions, offering remuneration for their sacrifices [61]. It is worth noting that compensating differentials can also operate inversely; lower wages may be offset by better working conditions.

**3.1.4 Discrimination theory.** Becker [62] identifies prejudice as the root cause of discrimination. He proposes that discrimination could be mitigated by monetization, introducing the market-based preference coefficient theory. This theory states that the preference coefficient equals the difference in the group wage rate, both when preference is present and when it is absent. Beyond gender, scholars have explored discrimination based on various other factors, including ethnicity [63, 64] and religion [65, 66]. Arrow, Ashenfelter and Rees [67] offer an

alternative viewpoint, suggesting that discrimination arises from incomplete information access and the attribution of group traits to individuals. This leads to the amplification of individual characteristics, which in turn leads to discrimination. Phelps [68] further refines this model of statistical discrimination, which is later adapted by Posner [69] to account for both inter-group and intra-group biases. Statistical discrimination compels job applicants to acquire skills that improve transparency for employers, thereby reducing discrimination [70–73].

### 3.2 Empirical study of the gender wage gap

**3.2.1 Human capital and the gender wage gap.** Labor economics scrutinizes the parity between the economic standing of men and women in the labor market, investigating whether the earnings of both groups are determined by identical mechanisms [74]. While numerous factors pertaining to human capital can contribute to the wage gap between men and women, the majority of studies concentrate on two dimensions: skill differential and skill return differential. The skill differential signifies the variances between men and women in aspects such as educational attainment and years of experience. Conversely, the skill return differential refers to disparities in the rate of return on education and length of service, among other variables [75, 76].

Some institutional reports and economic researchers posit that female workers possess lower levels of human capital in comparison to male workers [77–79]. Nonetheless, the literature exhibits inconsistencies concerning the return on human capital for both genders. For instance, while numerous studies have observed higher returns to education for women [80–82], other research indicates higher returns on human capital for men. Tverdostup and Paas [83] utilize the Program for International Assessment of Adult Competencies across 17 European countries and find that men are more likely to earn higher wages, despite generally possessing lower levels of formal education, owing to the presence of a "glass ceiling" for women. The existence of this glass ceiling is further corroborated by Harb and Rouhana [84], who apply counterfactual decomposition and generalized quantile regression in their study, which is based on Lebanese data. Their findings suggest that certain underlying elements, such as family responsibilities, adversely affect the return on human capital for women.

**3.2.2 Labor market segmentation and the gender wage gap.** Empirical inquiries into labor market segmentation first emerge in developed nations towards the end of the 20th century. Scholars subsequently assert that the gender wage gap is significantly influenced by sector segmentation, whether by occupation [85, 86], industry [87, 88], or degree of urbanization [17, 89]. Following extensive examination of wage gaps between the public and private sectors in developed countries, a consensus emerges about the prevalence of wage premiums in the public sector [90–94]. Shapiro and Stelcner [95] evidence this public sector wage premium utilizing Canadian census data and decompose the wage gap into endowment and residual differences. On the contrary, Dustmann and van Soest [96] report no such premiums in the public sector in Germany, where wages are markedly lower than in the private sector. Krueger [92] employs American panel data and finds that federal employees earn an average salary 10%–25% higher than their counterparts in the private sector. This finding is corroborated by Mueller [93] based on Canadian data, although such conclusions are not universally accepted.

However, the role that sector segmentation plays in the gender wage gap remains a subject of debate among scholars in developed countries. For instance, Gornick and Jacobs [97] argue that public employment has a limited impact on the overall gender wage gap in most nations. Yet, studies by Blau and Kahn [98] and Anner [99] indicate that occupational segmentation in the United States has seen a significant decline. Increasing academic focus has also been directed towards inter-sector wage gaps in developing countries, especially in Asia and Africa. For example, Clark et al. [100] apply Malaysian data to demonstrate higher wages in the public

sector and a decline in gender wage differentials, while Kwenda and Ntuli [101] observe an inverted U-shaped wage gap in the public and private sectors in South Africa.

In the Chinese context, researchers have analyzed the gender wage gap from various angles, including industry [102, 103], urban-rural discrepancies [17, 104, 105], party membership [12], and the Hukou system [106]. Notably, most of the research indicates that sector segmentation exacerbates the gender wage gap due to divergent wage determination mechanisms and historical factors. Several studies employing wage decomposition models have confirmed that employees in government and state-owned enterprises enjoy privileges [107–110]. Iwasaki and Ma [111] conduct a meta-analysis and conclude that the gender wage gap is more pronounced in rural and private sectors compared to urban areas and the public sector. Additionally, the implementation of the two-child policy since 2015 has spurred a growing number of Chinese studies to explore the intersectionality of fertility intentions and sectoral wage inequality [9, 112].

Nevertheless, the gender wage gap in China still needs further investigation, particularly focusing on sector segmentation. The existing literature often overlooks the role of the collective economy [102, 108, 113], and due to limitations in sample size and data availability, studies have yet to examine specific wage distributions concerning the gender wage gap and sector segmentation [114, 115]. Furthermore, many studies are constrained by data limitations when attempting to delineate the temporal trends of gender discrimination in ownership segmentation [116]. Finally, the existing literature uses cross-sectional data from several adjacent years and does not use proper decomposition methods to analyze the impact of intra-sector and inter-sector differences on the gender wage gap [4, 111, 117, 118].

Thus, this study aims to fill these gaps by focusing on gender wage differentials across wage groups in three sectors within the Chinese context, employing labor market segmentation theory to analyze their trends.

## 4 Data and methodology

### 4.1 Data

**4.1.1 Data sources.** The data for this study were sourced from two main repositories: the Urban Household Survey (UHS) conducted by the National Bureau of Statistics of China, and the China Trade Union Statistical Yearbook, compiled by the All-China Federation of Trade Unions. The UHS is a comprehensive survey covering households in four provinces, namely Shanghai, Liaoning, Sichuan, and Guangdong, which represent eastern, northeastern, western, and southern China respectively. The China Trade Union Statistical Yearbook, a nationally recognized source, ceased updates after 2013. This study selected three representative years— 2004, 2008, and 2013—to generate robust empirical findings. After data cleaning, the dataset included over 41,000 individual records from these years, encompassing key variables such as annual wages and sectors. Additional control variables like work experience, education, gender, marital status, ethnicity, occupation, and industry were also included. Due to inconsistencies in industry classification over the study period, the Industrial Classification and Codes for National Economic Activities (GB/T 4754–94) were employed for calibration for 15 sectors. The classification details can be checked in S1 Table, and the comprehensive definitions and descriptions of all variables are presented in Table 1.

**4.1.2 Descriptive statistics.** Table 2 shows descriptive statistics classified by gender for the years 2004, 2008, and 2013. From a sector perspective, the proportion of the public sector labor force fell from 55% in 2004 to less than 39% in 2013. In contrast, the number of people employed in the private sector increased significantly by about 20 percentage points, rising from 38% in 2004 to 58% in 2013. Between 2008 and 2013, the number of people employed in the private sector surpassed that of the public sector. Although the proportion of collective

**Table 1. Definition and description of variables.**

| | Variable Name | Description |
|---|---|---|
| Dependent variable | Wage | Annual wage, includes year-end bonus, subsidy, etc. |
| Explanatory variables | | |
| Individual | | |
| | Gender | Male = 1, Female = 0 |
| | Marital Status | Has Partner = 1, Others = 0 |
| | Ethnicity | Han = 1, Others = 0 |
| Human capital | | |
| | Education | Years of education |
| | Work experience | Start from of first job |
| Employment | | |
| | Sector | Public Sector, Private sector, and Collective Economy |
| | Occupation | Eight occupations |
| | Industry | Fifteen Industries |
| Province | | |
| | Province | Sichuan, Liaoning, Shanghai, Guangdong |

[a] Data were collected from China Urban Household Survey (2004, 2008, 2013)

economy employment in China's labor force decreased from 6.79% in 2004 to approximately 3% in 2013, it still played an important role in the Chinese economy in 2013. The proportion of men among public officials decreased over this period. In 2004, the proportion of male employees in the public sector was 14% higher than that of female employees, but by 2013, this gap had narrowed to 9%. Compared to male employees, the proportion of female employees in the private sector was approximately 10 percentage points higher in 2013 and showed a gradually increasing trend. In the collective economy sector, the proportion of female employees was slightly higher than that of male employees across all three periods.

Table 2 also displays data on individual characteristics beyond their respective sectors. Between 2004 and 2013, the length of work experience for men declined more significantly than that for women. During the same period, women surpassed men in educational attainment. With regard to ethnicity and marital status, there was no significant disparity between males and females. In terms of occupation, the proportion of the population engaged in agriculture and manufacturing declined significantly, while the proportion of people engaged in the service industry continued to rise. This shift can be attributed to a significant economic transformation from 2004 to 2013, characterized by sustained growth in the secondary and tertiary industries. In 2004, manufacturing and construction were the most popular job sectors for men. However, in 2013, they were replaced by office work, education, and scientific research. In 2013, the overall proportion of female employees engaged in clerical work and family and business services exceeded 60%. Generally speaking, there were more women working in the service sector during this period compared to men. However, their representation in manufacturing and business management positions was significantly lower than that of men.

Table 3 presents the wage conditions by gender and sector, along with corresponding T-tests. Overall, wages for both men and women across the three sectors witnessed an increase from 2004 to 2008. However, the public sector maintained a notably higher wage level compared to the collective economy and the private sector. Interestingly, although the collective economy and the private sector had similar wage levels, the collective economy had higher wages in 2004 and 2008 but lagged behind the private sector in 2013. From a gender

**Table 2. Descriptive statistics by gender.**

| | 2004 | | | 2008 | | | 2013 | | | All | | |
|---|---|---|---|---|---|---|---|---|---|---|---|---|
| | **Male** | **Female** | **Total** | **Male** | **Female** | **Total** | **Male** | **Female** | **Total** | **Male** | **Female** | **Total** |
| Sector (%) | | | | | | | | | | | | |
| Public sector | 61.49 | 47.56 | 55.27 | 44.96 | 37.36 | 41.66 | 42.87 | 34.07 | 38.99 | 49.26 | 39.38 | 44.91 |
| Private sector | 33.09 | 43.95 | 37.94 | 50.65 | 57.71 | 53.71 | 54.33 | 62.73 | 58.04 | 46.60 | 55.20 | 50.39 |
| Collective Enterprise | 5.41 | 8.49 | 6.79 | 4.39 | 4.93 | 4.63 | 2.80 | 3.20 | 2.97 | 4.14 | 5.43 | 4.71 |
| Edu | 11.903 | 11.848 | 11.878 | 12.083 | 12.112 | 12.096 | 12.473 | 12.552 | 12.508 | 12.165 | 12.185 | 12.174 |
| | (2.52) | (2.29) | (2.42) | (2.65) | (2.57) | (2.61) | (2.67) | (2.59) | (2.63) | (2.63) | (2.51) | (2.58) |
| Exp | 22.338 | 19.065 | 20.877 | 21.190 | 17.608 | 19.636 | 23.026 | 19.173 | 21.325 | 22.186 | 18.615 | 20.613 |
| | (10.68) | (9.95) | (10.48) | (10.67) | (9.52) | (10.34) | (11.24) | (10.10) | (10.92) | (10.90) | (9.89) | (10.62) |
| Ethnicity | 0.038 | 0.046 | 0.041 | 0.031 | 0.035 | 0.033 | 0.043 | 0.046 | 0.044 | 0.037 | 0.042 | 0.039 |
| (0 = Han 1 = Others) | (0.19) | (0.21) | (0.20) | (0.17) | (0.18) | (0.18) | (0.20) | (0.21) | (0.21) | (0.19) | (0.20) | (0.19) |
| Mar | 0.885 | 0.860 | 0.873 | 0.892 | 0.869 | 0.88 | 0.884 | 0.847 | 0.868 | 0.886 | 0.858 | 0.874 |
| (1 = Has partner 0 = Single) | (0.32) | (0.35) | (0.33) | (0.31) | (0.34) | (0.32) | (0.32) | (0.36) | (0.34) | (0.32) | (0.35) | (0.33) |
| Occupation (%) | | | | | | | | | | | | |
| Manager | 4.78 | 1.40 | 3.27 | 5.02 | 1.92 | 3.68 | 3.34 | 1.08 | 2.34 | 4.36 | 1.46 | 3.08 |
| Technique & Research | 17.11 | 16.26 | 16.73 | 22.75 | 19.27 | 21.24 | 23.70 | 18.45 | 21.38 | 21.37 | 18.04 | 19.90 |
| Clerks | 27.58 | 29.40 | 28.39 | 25.50 | 29.65 | 27.30 | 30.25 | 35.98 | 32.78 | 27.81 | 31.81 | 29.57 |
| House& Business Service | 6.29 | 11.74 | 8.72 | 17.89 | 31.68 | 23.88 | 17.59 | 31.97 | 23.94 | 14.25 | 25.57 | 19.24 |
| Agriculture | 10.60 | 24.11 | 16.63 | 0.62 | 0.41 | 0.53 | 0.35 | 0.19 | 0.28 | 3.56 | 7.72 | 5.39 |
| Production & Transport | 31.62 | 14.90 | 24.16 | 21.10 | 9.75 | 16.18 | 19.49 | 6.42 | 13.72 | 18.17 | 5.56 | 15.38 |
| Soldier | 0.20 | 0.09 | 0.50 | 0.69 | 0.18 | 0.47 | 0.61 | 0.05 | 0.36 | 0.07 | 4.72 | 7.72 |
| Others | 1.82 | 2.10 | 1.94 | 6.42 | 7.13 | 6.73 | 4.68 | 5.86 | 5.20 | 4.41 | 5.11 | 4.72 |
| Industry (%) | | | | | | | | | | | | |
| Agriculture | 1.02 | 0.71 | 0.88 | 1.10 | 0.61 | 0.89 | 1.00 | 0.63 | 0.84 | 1.04 | 0.65 | 0.87 |
| Mining | 1.85 | 0.55 | 1.27 | 2.26 | 1.22 | 1.81 | 2.13 | 0.66 | 1.48 | 2.09 | 0.81 | 1.53 |
| Manufacturing | 25.73 | 16.97 | 21.82 | 20.34 | 13.62 | 17.43 | 19.75 | 12.08 | 16.36 | 21.77 | 14.12 | 18.40 |
| Electricity, Gas and Water | 3.85 | 2.04 | 3.04 | 3.68 | 2.07 | 2.98 | 3.23 | 1.43 | 2.43 | 3.57 | 1.83 | 2.81 |
| Construction | 4.01 | 1.51 | 2.89 | 4.62 | 1.85 | 3.42 | 5.42 | 1.94 | 3.89 | 4.72 | 1.78 | 3.42 |
| Water and Environment | 1.40 | 1.10 | 1.27 | 1.22 | 0.81 | 1.04 | 0.99 | 0.81 | 0.91 | 1.19 | 0.90 | 1.07 |
| Transport and Information | 13.42 | 5.35 | 9.82 | 13.44 | 5.69 | 10.08 | 20.29 | 19.19 | 19.80 | 15.84 | 10.36 | 13.43 |
| Hotel and Restaurants | 13.05 | 19.51 | 15.93 | 13.70 | 23.69 | 18.04 | 8.42 | 10.78 | 9.46 | 11.64 | 17.82 | 14.36 |
| Financial Intermediation | 2.49 | 2.47 | 2.48 | 2.84 | 3.58 | 3.16 | 3.39 | 3.99 | 3.66 | 2.93 | 3.38 | 3.13 |
| Real Estate | 2.62 | 3.04 | 2.81 | 1.46 | 1.03 | 1.27 | 2.08 | 1.69 | 1.91 | 2.03 | 1.89 | 1.97 |
| House and Business Services | 8.52 | 19.12 | 13.25 | 13.23 | 19.48 | 15.95 | 10.75 | 18.80 | 14.30 | 10.92 | 19.13 | 14.54 |
| Health, sports and social welfare | 2.06 | 4.51 | 3.16 | 2.72 | 5.99 | 4.14 | 2.67 | 5.34 | 3.85 | 2.50 | 5.30 | 3.73 |
| Education, culture and broadcast | 6.30 | 8.78 | 7.41 | 5.79 | 9.09 | 7.22 | 5.39 | 9.21 | 7.07 | 5.80 | 9.03 | 7.23 |
| Scientific Research | 2.23 | 1.78 | 2.03 | 1.48 | 0.94 | 1.25 | 1.66 | 0.88 | 1.31 | 1.77 | 1.18 | 1.51 |
| Social Organization | 11.31 | 12.58 | 11.88 | 12.12 | 10.28 | 11.32 | 12.83 | 12.58 | 12.72 | 12.12 | 11.81 | 11.98 |
| Observations | 6,983 | 5,629 | 12,612 | 7,881 | 6,041 | 13,922 | 8,077 | 6,384 | 14,461 | 22,941 | 18,054 | 40,995 |

[a] In the UHS database, the degree of education is divided into seven categories: postgraduate, university, junior college, technical secondary school, high school, junior high school, and elementary school. According to China's education system, the corresponding education years are 18, 16, 14, 12, 12, 9, and 6.

[b] The UHS statistical division of industries has changed between these three survey years. This article re-divides the latest twenty industries into the original fifteen industries. The details can be found in Table A in S1 Table.

[c] Data were collected from China Urban Household Survey (2004, 2008, 2013)

**Table 3. Gender wage by sector.**

| | 2004 | | | 2008 | | | 2013 | | | All | | |
|---|---|---|---|---|---|---|---|---|---|---|---|---|
| | Public | Collective | Private | Public | Collective | Private | Public | Collective | Private | Public | Collective | Private |
| Male | 9.672 | 9.242 | 9.338 | 10.147 | 9.903 | 9.797 | 10.540 | 9.990 | 10.175 | 10.087 | 9.661 | 9.854 |
| *Std* | (0.010) | (0.032) | (0.017) | (0.011) | (0.036) | (0.013) | (0.012) | (0.058) | (0.013) | (0.007) | (0.025) | (0.009) |
| *Observation* | 4,292 | 378 | 2,304 | 3,543 | 346 | 3,992 | 3,463 | 226 | 4,388 | 11,298 | 950 | 10,684 |
| Female | 9.447 | 9.052 | 8.990 | 9.962 | 9.741 | 9.503 | 10.310 | 9.858 | 9.902 | 9.875 | 9.429 | 9.536 |
| *Std* | (0.014) | (0.031) | (0.016) | (0.014) | (0.041) | (0.013) | (0.016) | (0.063) | (0.014) | (0.009) | (0.026) | (0.009) |
| *Observation* | 2,677 | 478 | 2,474 | 2,257 | 298 | 3,484 | 2,175 | 204 | 4,005 | 7,109 | 980 | 9,963 |
| Wage Diff | 0.224 | 0.190 | 0.348 | 0.184 | 0.161 | 0.293 | 0.229 | 0.132 | 0.273 | 0.212 | 0.231 | 0.317 |
| T-test | -13.04 | -4.10 | -14.81 | -10.07 | -2.93 | -15.65 | -11.02 | -1.52 | -13.70 | -17.54 | -6.21 | -24.83 |
| P-value | 0.000 | 0.000 | 0.000 | 0.000 | 0.003 | 0.000 | 0.000 | 0.127 | 0.000 | 0.000 | 0.000 | 0.000 |

[a] Data were collected from China Urban Household Survey (2004, 2008, 2013)

perspective, men consistently earned more than women across all sectors, the disparity being particularly stark in the private sector. Conversely, the gender wage gaps were relatively narrower in the collective economy. Paired samples T-tests were conducted for gender wages in various sectors and years, confirming the existence of significant wage gaps between males and females.

Fig 1 displays the wage distribution of male and female employees in different sectors in 2004, 2008, and 2013 according to a kernel density estimation. In 2004, the wage distribution of male and female employees in the public sector was quite similar. However, the wages of male employees in the private sector and collective economy were significantly higher than those of female employees, and the wage level of the latter was more concentrated. In 2008, the wages of female employees in the public sector also began to gradually lag behind those of male employees, showing a trend similar to that of the private sector and the collective economy. This indicates that the wage gap between men and women began to widen in all sectors. By 2013, the average and peak levels of wages for male employees in the public sector were higher than those of female employees, while the peak wages of male employees in the private sector were roughly the same, although their overall wage gap was smaller. It is worth noting that in 2004 and 2008, the wage levels of male employees in the collective economy were higher than those of female employees. However, by 2013, the wage distribution curves for both genders almost coincided, which may be related to the reduction in the number of workers in this sector and technological advancements in agriculture. Such advancements may have reduced the impact of physical strength on wages, leading to a reduction in the gender wage gap.

Fig 2 delineates the wage distribution of men and women across different wage percentiles and their respective gender wage ratios. The left vertical axis signifies the logarithmic values of annual wages, while the right vertical axis represents the male-female wage ratio. The horizontal axis corresponds to different wage percentiles. Macroscopically, the absolute value of the gender wage gap gradually widened over this period, although the gender wage ratio remained stable. Male-female wage ratios varied considerably across different time periods and sectors. In 2004, in all three sectors, gender wage ratios decreased with rising wage percentiles. However, in 2008 and in 2013, the gender wage ratio displayed a "high at both ends, low in the middle" curve for both the private sector and the collective economy. Put differently, the gender wage gap was more pronounced among high and low-wage groups compared to middle-wage groups. The pattern in 2013 for the public sector mirrored that of 2004. Overall, gender wage

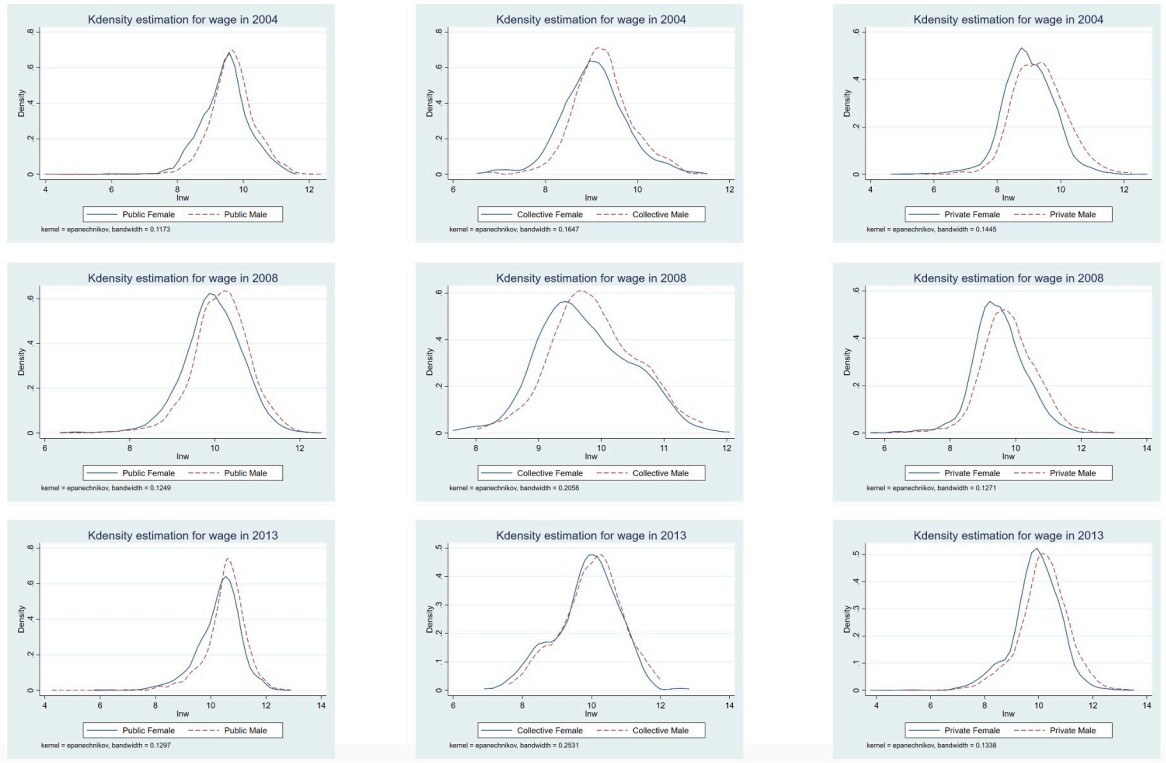

**Fig 1. Kernel density distribution of gender wages by sector and year.**

gaps during this period were most evident among low-wage groups in the public sector, private sector and collective economy, but also evident among high-wage groups.

## 4.2 Empirical methods

**4.2.1 Wage function.** To estimate the impact of sector segmentation on male and female wages, the study first employs a basic Ordinary Least Squares (OLS) model, formalized as Eq (4.1) and grounded on the Mincer equation [58]:

$$lnw_i = \beta_0 + \beta_1 \, Sector_i + \beta_x X_i + u_i, \tag{4.1}$$

In this context, $lnw_i$ stands for the logarithmic form of annual wage. Annual wages are taken in logarithmic form because this makes the data more stationary and is also conducive to explaining the effects of the independent variables more easily [119]. *Sector* refers to the public sector, private sector, and collective economy; and $X_i$ designates other control variables, including individual attributes such as gender, ethnicity, marital status, and province, as mentioned in Table 1. It also includes human capital features like education and work experience, and job attributes such as occupation and industry. $\beta_x$ in this instance denotes the wage premium associated with a specific sector.

**4.2.2 Bourguignon, Fournier and Gurgand model.** Traditional techniques to address the issue of self-selectivity, such as the Heckman sample selection model and the treatment effects model, are inherently limited to bivariate cases. These models cannot be directly applied when the treatment variable is multivariate [120], as in this study where the variables include the public sector, private sector, and collective economy.

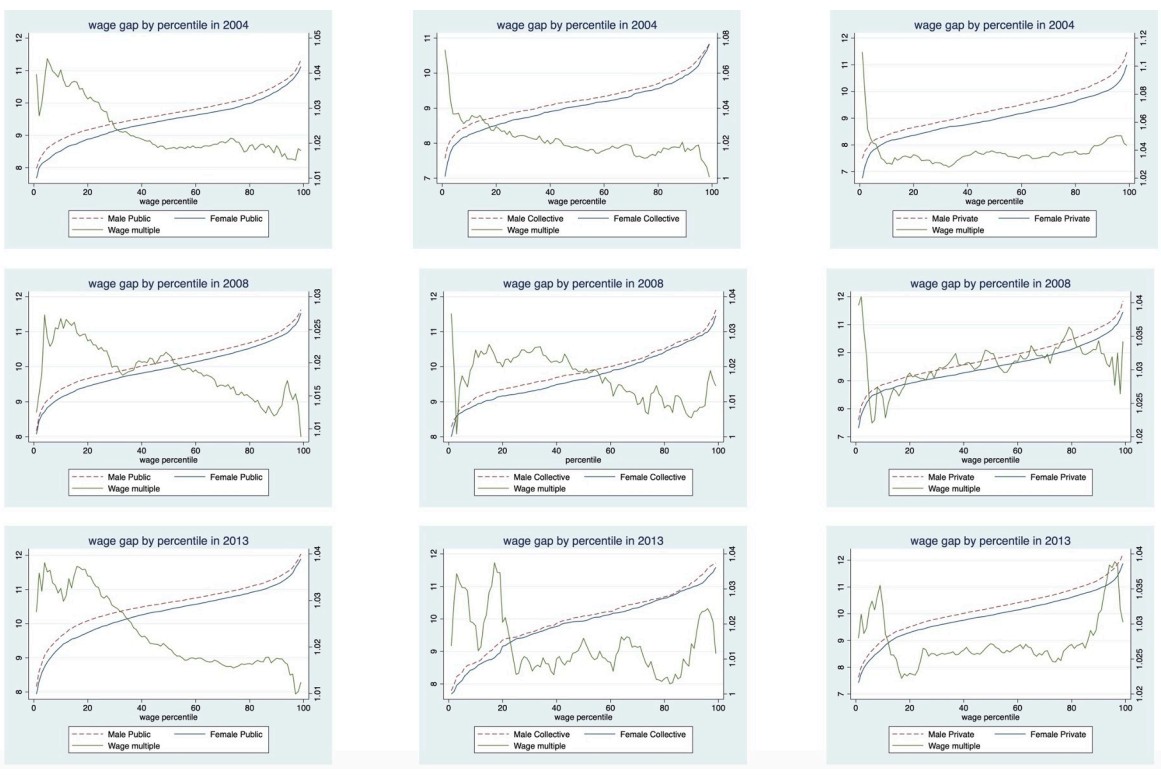

**Fig 2. Wage gaps by gender, sector and year.**

This study uses the Bourguignon, Fournier and Gurgand model [121]. This model accommodates a polychotomous selection process, thereby allowing for multiple categories. The methodology comprises a two-step generalized approach that can incorporate OLS computations:

$$y_s = x_s \beta'_s + u_s, \tag{4.2.1}$$

Here, the model assumes a categorical variable $S = 1, \ldots, M$ (more than two categories) that represents choices based on individual utilities as:

$$y_s^* = z_s \gamma_s + \eta_s, \tag{4.2.2}$$

where $z_s$ and $\eta_s$ compose a vector of independent variables and the disturbance term which confirms the usual conditions. The impact on the dependent variable is observed only for the case in which the alternative $S$ is chosen:

$$y_s^* > \max_{j \neq s} \left( y_j^* \right) \tag{4.2.3}$$

$$\varepsilon_s = \max_{j \neq s} \left( y_j^* - \eta_s \right); \varepsilon_s < 0, \tag{4.2.4}$$

Upon calculating cumulative and density functions [122], the multinomial logit specification is employed:

$$P(z_s\gamma_s > \varepsilon_s) = \frac{\exp(z_s\gamma_s)}{\sum_j \exp\left(z_j\gamma_j\right)} \tag{4.2.5}$$

$$\ln y_s = \beta'_s x_s + \varepsilon_s - \sigma_{\eta u}\rho'_s, \tag{4.2.6}$$

where $\sigma_{\eta u}\rho'_s$ are coefficient terms for the polychotomous correction of selectivity bias.

**4.2.3 Recentered influence function regression.** This method above is confined to mean analysis, which inhibits an in-depth examination of the wage distribution. Moreover, the wage distribution among employees across sectors may be skewed; for example, the private sector may exhibit more severe wage polarization compared to the public sector, based on mean differences. Hence the recentered influence function (RIF) regression, devised by Firpo, Fortin and Lemieux [123], is employed to delve into the impact of sector segmentation on wage gaps and to identify which characteristics contribute to gender wage inequality. The RIF model reconfigures distribution statistics to enable more precise regression analyses. Therefore, the RIF quantile regression has merit as a comprehensive depiction of the wage distribution across each quantile. By decomposing the wage gaps across sectors into characteristic and coefficient effects, the contribution of each explanatory variable can be quantified. Mathematically, RIF is represented as:

$$RIF(Y; v) = v(F_Y) + IF(Y; v), \tag{4.3.1}$$

where $v$ represents various statistics describing the distribution of $F_Y$; and $IF(Y; v)$ is the influence function corresponding to the specific statistic $Y$. When the distribution statistic is quantile, RIF regression belongs to unconditional quantile regression. The RIF of the $Y$ variable at the $Q_t$ quantile can be expressed as:

$$RIF(Y; v) = Q_t + \frac{\tau - \{Y \leqslant Q_t\}}{f_Y(Q_t)}, \tag{4.3.2}$$

where $f_Y$ is the marginal density function of $Y$; $Q_t$ is the unconditional distribution of $t$ quantiles; and $RIF(Y; v)$ is a function that can linearly represent other explained variables. Additionally, in analyzing the influence of variables such as sectors on the wages of different quantiles of each sample, the following equation can be constructed for the unconditional quantile regression:

$$RIF(lnw; Q_r) = X_i\beta_i + \varepsilon, \tag{4.3.3}$$

where $Q_r$ is the quantile of wages; and $X_i$ represents variables such as human capital and work characteristics.

**4.2.4 Brown decomposition.** Since the gender wage gap is the result of a combination of inter-sectoral and intra-sectoral differentials, a more elaborate decomposition of the wage gap is warranted. The Brown decomposition model [124] is a sound approach which can be adapted to compare the impact of intra- and inter-sectoral variables on the gender wage gap using percentage values. When using this model, imputed probabilities of entering sectors are estimated using a multinomial logit regression model, accounting for sample selection bias [125]:

$$\ln W_{iK} = \alpha_K + \beta_{KX}X_{iK} + \beta_{K\delta}\delta_{iK} + u_{iK}, \tag{4.4.1}$$

The Probit regression model is used in which $P_{ik} = \text{Prob}(y_{ik} = \text{Sector}_{ik})$ to indicate the probability of entry to one sector. The selectivity items $(\delta = \psi(\cdot)/\Phi(\cdot))$ for various ownership

types are calculated. The decomposition contents can be expressed as:

$$\ln W_m - \ln W_f = \sum P_k^f \beta_k^m \left( X_k^m - X_k^f \right) \text{(A)}$$
$$+ \sum P_k^f \left( \alpha_k^m - \alpha_k^f \right) + \sum P_k^f X_k^f \left( \beta_k^m - \beta_k^f \right) \text{(B)}$$
$$+ \sum W_k^m \left( P_k^m - P_k^{*f} \right) \text{(C)}$$
$$+ \sum W_k^u \left( P_k^{*f} - P_k^m \right) \text{(D)}$$

$$(4.4.2)$$

where $P_k^f$ and $P_k^f$ represent the actual proportions of female and male groups, $\hat{P}_k^f$ represents the imputed proportions of the female group, $X_k^m$ and $X_k^f$ represent mean values of variables, and $\beta_k^m$ and $\beta_k^f$ are the parameters estimated based on wage functions by sector categories. Furthermore, (A) represents the individual characteristic differentials between male and female groups in a given sector (the explained component in intra-sector differentials); (B) represents the unexplained component (discrimination against female workers in the same sector) in a given sector (the unexplained component in intra-sector differentials); (C) represents the individual characteristic differentials between male and female workers which determine the chance (probability) of entry to various ownership sectors (the explained component in inter-sector differentials); and (D) represents the unexplained component (discrimination against female workers) when they enter a sector (the unexplained component in inter-sector differentials).

Here, (A) and (B) capture the total intra-sector differential, while (C) and (D) encapsulate the total inter-sector differential. (B) and (D) signify the total unexplained differential due to discrimination when female workers enter a sector or work alongside male workers in the same sector. (A) and (C) capture the total explained differential.

## 5 Empirical research results

### 5.1 BFG model results

Table 4 shows the estimated results of the wage function by sector. Overall, the regression coefficients for gender were highly significant. There was a wage premium for males in all three sectors, but the premiums varied. The gender wage premium was primarily concentrated in the private sector, which became the sector with the largest gender wage gap in 2013. The gender wage gap in the public sector was found to have diminished and stabilized at a relatively low level. This is consistent with the findings from the descriptive statistical analyses. The collective economy, however, underwent a transition as the gender wage gap reduced substantially. In 2004, male employees in the public sector earned wages that were 19.5% higher than those of female employees. In the private sector, the difference was nearly 23.4%. Notably, the wage gap in the collective economy was 38.1%. In 2008, the collective economy maintained the widest gender wage gap at 22.8%, followed by the private sector at 14.2%. The public sector also recorded a wage gap of 14.2%. By 2013, the private sector had overtaken the collective economy, registering the largest gender wage gap of 29.3%. The gender wage gap in the collective economy stood at 13.1%, while the public sector had the smallest gap, at 10.2%. Based on data from these three years, policies aimed at reducing the gender wage gap were most effective in the public sector and the collective economy, whereas the wage gap in the private sector actually widened.

Work experience and education were other control variables which had relatively large impacts on the wage gap. Work experience played an important role in influencing wages, especially in the collective economy and the public sector. For example, from 2004 to 2013, for every additional year of work experience, wages in the public sector increased by about 3%. In

**Table 4. Estimated results of wage function by sector.**

|  |  | 2004 |  |  | 2008 |  |  | 2013 |  |
|---|---|---|---|---|---|---|---|---|---|
|  |  | (1) | (2) | (3) | (1) | (2) | (3) | (1) | (2) | (3) |
|  |  | Public | Collective | Private | Public | Collective | Private | Public | Collective | Private |
| Male | 0.195*** | 0.381*** | 0.234*** | 0.108*** | 0.228*** | 0.142* | 0.102** | 0.131 | 0.293*** |
|  | (0.0518) | (0.115) | (0.0400) | (0.0322) | (0.0729) | (0.0765) | (0.0436) | (0.216) | (0.0509) |
| Exp | 0.0268*** | 0.0479* | 0.0220*** | 0.0330*** | 0.0206 | 0.0172* | 0.0283*** | 0.0427 | 0.0143 |
|  | (0.00974) | (0.0291) | (0.00602) | (0.00828) | (0.0177) | (0.00951) | (0.00830) | (0.0545) | (0.0127) |
| Exp2 | -0.0004* | -0.00108 | -0.000187 | -0.000492** | -0.000472 | -0.000178 | -0.000421** | -0.000703 | -0.000354 |
|  | (0.000225) | (0.000676) | (0.000131) | (0.000199) | (0.000396) | (0.000184) | (0.000199) | (0.00103) | (0.000224) |
| Edu | 0.134*** | 0.0764*** | 0.112*** | 0.176*** | 0.0644*** | 0.165*** | 0.277*** | 0.105*** | 0.160*** |
|  | (-0.005) | (0.007) | (0.005) | (-0.005) | (-0.010) | (0.004) | (-0.072) | (-0.019) | (-0.004) |
| Ethnicity | -0.00337 | -0.0618 | 0.0415 | 0.0320 | -0.0453 | -0.146* | 0.0862 | -0.278 | -0.0802 |
|  | (0.0586) | (0.203) | (0.0614) | (0.0700) | (0.177) | (0.0774) | (0.0694) | (0.409) | (0.0991) |
| Partner | -0.385 | -2.923 | -10.77* | 1.534 | -23.10 | -1.995 | 1.602 | -2.493 | 0.633 |
|  | (1.317) | (34.79) | (6.430) | (77.62) | (34.46) | (10.91) | (14.35) | (67.39) | (2.702) |
| Technique | 0.0753* | 0.236 | 0.295*** | -0.162*** | 0.187 | -0.00343 | -0.123** | 1.268 | -0.747*** |
|  | (0.0390) | (0.358) | (0.109) | (0.0601) | (0.309) | (0.286) | (0.0558) | (6.304) | (0.166) |
| Clerks | -0.0843** | -0.396** | 0.0462 | -0.206*** | 0.220 | -0.183 | -0.273*** | 1.005 | -0.842*** |
|  | (0.0328) | (0.166) | (0.0530) | (0.0673) | (0.308) | (0.236) | (0.0551) | (6.268) | (0.175) |
| Service | -0.361*** | -0.507** | -0.249*** | -0.378*** | -0.00282 | -0.269 | -0.271** | 1.605 | -0.879*** |
|  | (0.0984) | (0.240) | (0.0557) | (0.133) | (0.314) | (0.276) | (0.120) | (6.320) | (0.179) |
| Agriculture | -0.364*** | -0.490*** | -0.346*** | -0.288* | 0.478 | -0.433 | - | - | -1.495*** |
|  | (0.0490) | (0.176) | (0.0459) | (0.165) | (0.416) | (0.345) | - | - | (0.480) |
| Production | -0.165*** | -0.639*** | -0.357*** | -0.341*** | -0.0138 | -0.477* | -0.304** | 1.643 | -0.997*** |
|  | (0.0522) | (0.202) | (0.0506) | (0.0914) | (0.314) | (0.272) | (0.133) | (6.298) | (0.180) |
| Soldier | -0.0657 | - | -1.294*** | 0.0185 | - | - | 0.232** | - | -1.259** |
|  | (0.287) | - | (0.370) | (0.0988) | - | - | (0.100) | - | (0.498) |
| Others | -0.334* | -0.544 | -0.488*** | -0.295 | -0.167 | -0.291 | -0.0254 | 1.090 | -1.020*** |
|  | (0.196) | (0.390) | (0.125) | (0.219) | (0.400) | (0.287) | (0.273) | (6.330) | (0.167) |
| Shanghai | 0.655*** | 0.453*** | 0.707*** | 0.715*** | 0.396** | 0.966*** | 0.713*** | 1.076 | 0.689*** |
|  | (0.0843) | (0.157) | (0.0550) | (0.0851) | (0.198) | (0.107) | (0.0867) | (0.887) | (0.0753) |
| Guangdong | 0.559*** | 0.548*** | 0.551*** | 0.437*** | 0.462*** | 0.523*** | -0.0599 | -0.155 | -0.152* |
|  | (0.0323) | (0.1000) | (0.0362) | (0.0529) | (0.120) | (0.0816) | (0.0513) | (0.333) | (0.0879) |
| Sichuan | 0.0534 | -0.0283 | -0.0251 | -0.0774*** | -0.0775 | 0.0372 | 0.0455 | -0.0130 | 0.0182 |
|  | (0.0332) | (0.129) | (0.0478) | (0.0300) | (0.0930) | (0.0440) | (0.0419) | (0.223) | (0.0569) |
| M1 | 1.612 | 2.866 | -17.86 | 8.603 | -572.7 | 2.302 | 19.37 | -24.99 | -0.628 |
|  | (2.457) | (129.2) | (12.58) | (364.8) | (645.1) | (28.91) | (37.00) | (347.8) | (6.901) |
| M2 | -2.354 | -7.954 | -29.06** | -139.2 | 569.0 | -174.5 | -248.7 | 0 | 31.04 |
|  | (19.13) | (206.7) | (13.98) | (4,725) | (719.0) | (689.6) | (417.6) | (344.5) | (142.4) |
| M3 | 6.532 | 22.92 | 32.43* | 19.91 | -741.4 | 20.00 | 10.56 | 1.298 | -0.0249 |
|  | (13.73) | (298.1) | (19.40) | (526.3) | (941.3) | (78.34) | (64.63) | (393.2) | (10.39) |
| Constant | 11.17 | 37.45 | 13.39 | -8.764 | -1,595 | -2.123 | -42.04 | 13.67 | 11.44*** |
|  | (9.438) | (496.0) | (8.455) | (873.2) | (1,989) | (46.12) | (139.0) | (1,083) | (3.960) |
| Ancillary |  |  |  |  |  |  |  |  |  |
| Sigma2 | 4.431 | 69.29 | 11.44 | 297.3 | 108.3 | 35.72 | 204.0 | 1,436 | 10.46 |
|  | (4.458) | (162.0) | (7.424) | (624.6) | (29,794) | (784.0) | (586.8) | (15,888) | (322.0) |
| rho1 | 0.766 | 0.344 | -5.280 | 0.499 | -55.03*** | 0.385 | 1.356 | -0.659 | -0.194 |
|  | (0.914) | (20.29) | (3.768) | (9.117) | (4.979) | (1.506) | (3.553) | (62.19) | (1.430) |

(*Continued*)

**Table 4.** (Continued)

|  | | 2004 | | | 2008 | | | 2013 | |
|---|---|---|---|---|---|---|---|---|---|
| rho2 | -1.118 | -0.956 | -8.590** | -8.072 | 54.67*** | -29.19 | -17.41 | 0 | 9.597 |
|  | (5.493) | (31.57) | (3.975) | (156.9) | (4.948) | (36.09) | (54.15) | (62.98) | (13.79) |
| rho3 | 3.103 | 2.753 | 9.587* | 1.155 | -71.23*** | 3.347 | 0.740 | 0.0342 | -0.00771 |
|  | (5.454) | (46.21) | (5.692) | (24.31) | (6.595) | (4.121) | (5.530) | (71.69) | (1.322) |
| Observations | N = 12,603 | | | N = 13,920 | | | N = 14,461 | | |

[a] Due to space constraints, the tables do not present regression results on industries

[b] Standard error in parentheses.

* $p < 0.1$

** $p < 0.05$

*** $p < 0.01$

[c] Data were collected from China Urban Household Survey (2004, 2008, 2013)

the collective economy, each additional year of experience led to a wage increase of over 4%, whereas in the private sector, the increase was approximately 2%. From a human capital standpoint, education has increasingly become a crucial factor. In the public sector, each additional year of education contributed to a wage increase of 13.4% in 2004, but this had risen to 27.7% by 2013. Meanwhile, the private sector saw an increase from 11.2% to 16.0%. In the collective economy, education had a relatively lower impact—around 10%—although the coefficient was still statistically significant. It is noteworthy that the regression coefficient concerning the primary ethnic group (Han ethnicity) was rather small and statistically insignificant. This suggests that there was no overt wage discrimination against ethnic minorities. The influence of marital status on wages across different sectors was also not significant.

## 5.2 RIF regression results

This study further employed the RIF quantile regression method to investigate the influence of various sectors on wage gaps. Tables 5–7 present the regression coefficients at the 10th, 50th, and 90th quantiles. Most coefficients were found to be statistically significant. Broadly speaking, the gender wage gap in the public sector diminished with rising wages. Conversely, in the private sector, the gender wage gap among wealthier cohorts tended to expand rapidly as wages increased. In the collective economy, the gender wage gap remained relatively stable with increasing wages. In 2004, within the public sector, men's wages were 33.9% higher than women's at the 10th quantile, and 17.5% higher at the 90th quantile. The gender wage variations in the public sector in the years 2008 and 2013 were generally in line with those of 2004. In the private sector, wage gaps among lower-wage groups were slightly smaller. For instance, in 2004, men earned 24% more than women at the 10th quantile, but this figure escalated swiftly to 39% within the 90th quantile.

Overall, the three-year regression results showed that in the private sector, the higher the wage level, the greater the gender wage gap. The collective economy maintained a relatively stable gender wage gap over these years with increasing wages. In the early stages of the collective economy, gender inequality at each quantile was quite pronounced—for example, in 2004, the gender wage gap was as high as 43.4% at the 10th quantile, but in 2008 and 2013, it remained consistently around 20% at varying quantiles. A possible reason for this result is that in the early stage of the collective economy, which was dominated by labor-intensive agriculture and handicrafts, males had a natural advantage over females. However, with the

**Table 5. RIF quantile regression by sector in 2004.**

|  | 10% | | | 50% | | | 90% | | |
|---|---|---|---|---|---|---|---|---|---|
|  | **Public** | **Collective** | **Private** | **Public** | **Collective** | **Private** | **Public** | **Collective** | **Private** |
| Gender | 0.339*** | 0.434*** | 0.243*** | 0.175*** | 0.304*** | 0.246*** | 0.175*** | 0.263*** | 0.394*** |
|  | (0.035) | (0.083) | (0.032) | (0.017) | (0.050) | (0.029) | (0.034) | (0.094) | (0.042) |
| Exp | 0.048*** | 0.034* | 0.035*** | 0.026*** | 0.036*** | 0.016*** | 0.012* | 0.017 | 0.017*** |
|  | (0.009) | (0.020) | (0.006) | (0.004) | (0.011) | (0.005) | (0.007) | (0.024) | (0.006) |
| Exp2 | -0.001*** | -0.001* | -0.000*** | -0.000*** | -0.001** | 0.000* | -0.000 | -0.000 | -0.000* |
|  | (0.000) | (0.000) | (0.000) | (0.000) | (0.000) | (0.000) | (0.000) | (0.001) | (0.000) |
| Edu | 0.085*** | 0.040** | 0.051*** | 0.074*** | 0.057*** | 0.105*** | 0.104*** | 0.109*** | 0.160*** |
|  | (0.009) | (0.018) | (0.007) | (0.004) | (0.012) | (0.006) | (0.009) | (0.025) | (0.010) |
| Ethnicity | 0.059 | 0.262 | 0.068 | 0.228*** | 0.215** | 0.128* | 0.457*** | 0.240* | 0.239*** |
|  | (0.077) | (0.223) | (0.094) | (0.039) | (0.100) | (0.074) | (0.039) | (0.138) | (0.073) |
| Partner | 0.063 | 0.052 | -0.087 | 0.099*** | -0.082 | 0.009 | 0.097 | -0.119 | 0.175*** |
|  | (0.072) | (0.163) | (0.057) | (0.033) | (0.103) | (0.048) | (0.060) | (0.193) | (0.061) |
| Cons | 6.859*** | 7.262*** | 7.412*** | 7.863*** | 7.607*** | 7.995*** | 8.194*** | 7.956*** | 8.371*** |
|  | (0.187) | (0.407) | (0.166) | (0.084) | (0.215) | (0.146) | (0.163) | (0.450) | (0.194) |
| Observations | 6,969 | 856 | 4,778 | 6,969 | 856 | 4,778 | 6,969 | 856 | 4,778 |

[a] Due to space constraints, the tables do not present regression results on industries, occupations, and provinces

[b] Standard errors in parentheses.

* $p < 0.1$

** $p < 0.05$

*** $p < 0.01$

[c] Data were collected from China Urban Household Survey (2004)

modernization of agriculture and the popularization of agricultural science and technology in China, the influence of physical gender factors in agricultural production gradually weakened.

### 5.3 Brown decomposition results

To delve deeper into the factors influencing the gender wage gap, particularly the discriminatory practices faced by female workers both when entering a sector and within a sector, the Brown decomposition method was employed. The outcomes are presented in Table 8.

Firstly, the influence of inter-sector differentials significantly outweighed that of intra-sector differentials across all three years examined. Inter-sector differentials accounted for nearly 90% of the total wage differentials and remained stable from 2004 to 2013. In essence, the results suggest that inter-sector differentials were the predominant factor driving the gender wage gap during this period.

Secondly, when assessing the cumulative effects of both explained and unexplained differentials, the influence of explained differentials in 2004 stood at 36.9%, markedly lower than that of the unexplained differentials. This trend remained consistent throughout the period, indicating that discrimination against female workers had a greater impact than labor endowment variables like human capital, across all three years. The findings also underscore the persistent nature of this inequality.

Thirdly, the unexplained component of the inter-sector differentials scored the highest in our overall decomposition results. These findings highlight discrimination against female workers within the same sector as the primary cause of the gender wage gap across these years. Notably, the influence of this component surged from 63.52% in 2002 to 77.99% in 2013.

**Table 6. RIF quantile regression by sector in 2008.**

|  | 10% | | | 50% | | | 90% | | |
|---|---|---|---|---|---|---|---|---|---|
|  | **Public** | **Collective** | **Private** | **Public** | **Collective** | **Private** | **Public** | **Collective** | **Private** |
| Gender | 0.275*** | 0.195** | 0.199*** | 0.209*** | 0.285*** | 0.260*** | 0.165*** | 0.180* | 0.309*** |
|  | (0.040) | (0.081) | (0.028) | (0.021) | (0.067) | (0.022) | (0.028) | (0.093) | (0.034) |
| Exp | 0.046*** | 0.020 | 0.014*** | 0.026*** | 0.008 | 0.018*** | 0.016*** | 0.006 | 0.020*** |
|  | (0.009) | (0.017) | (0.005) | (0.004) | (0.014) | (0.004) | (0.005) | (0.021) | (0.006) |
| Exp2 | -0.001*** | -0.000 | -0.000** | -0.000*** | -0.000 | -0.000*** | -0.000** | -0.000 | -0.000*** |
|  | (0.000) | (0.000) | (0.000) | (0.000) | (0.000) | (0.000) | (0.000) | (0.001) | (0.000) |
| Edu | 0.097*** | 0.032* | 0.060*** | 0.086*** | 0.094*** | 0.095*** | 0.074*** | 0.093*** | 0.138*** |
|  | (0.010) | (0.019) | (0.006) | (0.005) | (0.015) | (0.004) | (0.007) | (0.022) | (0.008) |
| Ethnicity | -0.105 | 0.221 | -0.125 | -0.047 | 0.224 | 0.076 | 0.262*** | 0.262*** | 0.068 |
|  | (0.081) | (0.280) | (0.076) | (0.055) | (0.146) | (0.062) | (0.051) | (0.101) | (0.086) |
| Partner | 0.239*** | -0.157 | 0.180*** | 0.154*** | 0.240** | 0.169*** | 0.169*** | 0.082 | 0.273*** |
|  | (0.087) | (0.127) | (0.050) | (0.040) | (0.107) | (0.034) | (0.045) | (0.155) | (0.050) |
| Cons | 7.294*** | 8.323*** | 8.020*** | 8.545*** | 7.972*** | 8.181*** | 9.398*** | 8.921*** | 8.786*** |
|  | (0.208) | (0.505) | (0.141) | (0.106) | (0.327) | (0.102) | (0.130) | (0.401) | (0.156) |
| Observations | 5,800 | 644 | 7,476 | 5,800 | 644 | 7,476 | 5,800 | 644 | 7,476 |

[a] Due to space constraints, the tables do not present regression results on industries, occupations, and provinces

[b] Standard errors in parentheses.

* p < 0.1

** p < 0.05

*** p < 0.01

[c] Data were collected from China Urban Household Survey (2008)

Conversely, the low and negative values of the unexplained components indicate that intra-sector differentials were less consequential.

Lastly, individual characteristics such as human capital and sector differentials also played a role, albeit a minor one, in the gender wage gap. When looking at explained and unexplained components within both inter-sector and intra-sector differentials, the explained components had less impact on the inter-sector differentials while being the key factor in the intra-sector differentials.

## 5.4 Results discussion

Firstly, judging from the overall gender wage gap, there is noticeable sector segmentation in the Chinese labor market. The gender wage gap in the public sector remained stable and low during this period, while the private sector replaced the collective economy as the sector with the largest gender wage gap. Private sector organizations are typically focused on generating profit, with the aim of creating value for their shareholders. Gender discrimination can manifest in various ways, such as lower wages, limited promotion prospects, and unfair working conditions for female employees. While the public sector and collective economy may also face gender discrimination issues, they often implement measures to minimize the gender wage gap. For instance, the public sector establishes fair wage systems, adopts policies and plans that promote gender equality, and ensures equal opportunities for promotion. On the other hand, the collective economy operates on cooperative principles, with decisions typically made collectively by members, reducing the likelihood of gender discrimination [126].

**Table 7. RIF quantile regression by sector in 2013.**

|  | 10% | | | 50% | | | 90% | | |
|---|---|---|---|---|---|---|---|---|---|
|  | **Public** | **Collective** | **Private** | **Public** | **Collective** | **Private** | **Public** | **Collective** | **Private** |
| Gender | 0.239*** | 0.241 | 0.316*** | 0.199*** | 0.232** | 0.249*** | 0.189*** | 0.261* | 0.261* |
|  | (0.053) | (0.188) | (0.057) | (0.019) | (0.105) | (0.022) | (0.029) | (0.134) | (0.134) |
| Exp | 0.004 | 0.005 | 0.011 | 0.031*** | 0.021 | 0.011*** | 0.021*** | 0.033 | 0.033 |
|  | (0.011) | (0.037) | (0.010) | (0.004) | (0.022) | (0.004) | (0.006) | (0.029) | (0.029) |
| Exp2 | 0.000 | -0.000 | -0.000* | -0.000*** | -0.001 | -0.000** | -0.000*** | -0.001 | -0.001 |
|  | (0.000) | (0.001) | (0.000) | (0.000) | (0.000) | (0.000) | (0.000) | (0.001) | (0.001) |
| Edu | 0.097*** | 0.075* | 0.055*** | 0.099*** | 0.095*** | 0.088*** | 0.097*** | 0.153*** | 0.153*** |
|  | (0.013) | (0.039) | (0.011) | (0.004) | (0.023) | (0.004) | (0.007) | (0.031) | (0.031) |
| Ethnicity | -0.126 | -0.450*** | -0.350*** | 0.021 | 0.331 | -0.032 | 0.220*** | -0.122 | -0.122 |
|  | (0.093) | (0.162) | (0.103) | (0.041) | (0.309) | (0.054) | (0.050) | (0.405) | (0.405) |
| Partner | 0.186* | 0.249 | 0.393*** | 0.106*** | 0.195 | 0.141*** | 0.148*** | 0.143 | 0.143 |
|  | (0.096) | (0.352) | (0.089) | (0.035) | (0.196) | (0.033) | (0.044) | (0.283) | (0.283) |
| Cons | 8.238*** | 8.014*** | 8.476*** | 8.851*** | 8.128*** | 9.051*** | 9.472*** | 9.147*** | 9.147*** |
|  | (0.245) | (0.770) | (0.250) | (0.091) | (0.549) | (0.099) | (0.147) | (0.656) | (0.656) |
| Observations | 5,638 | 430 | 8,393 | 5,638 | 430 | 8,393 | 5,638 | 430 | 430 |

[a] Due to space constraints, the tables do not present regression results on industries, occupations, and provinces

[b] Standard errors in parentheses.

* p < 0.1

** p < 0.05

*** p < 0.01

[c] Data were collected from China Urban Household Survey (2013)

Secondly, distinct wage groups within different sectors also exhibit significant variations in the gender wage gap. Specifically, the public sector sees a constant reduction in the gender wage gap as wages rise. In contrast, the private sector features more pronounced gender wage gaps among both low-wage and high-wage groups. The collective economy exhibited a considerable gender wage gap among low-wage individuals in 2004, but more recently it has demonstrated a balanced pattern across different wage quantiles. Notably, the gender wage gap is significant among low-wage individuals across all sectors. This could be attributed to women

**Table 8. Results based on the Brown decomposition.**

|  | 2004 | | 2008 | | 2013 | |
|---|---|---|---|---|---|---|
|  | **Actual value** | **Percentage** | **Actual value** | **Percentage** | **Actual value** | **Percentage** |
| Total wage differentials | 0.3259 | 100.00% | 0.2885 | 100.00% | 0.3009 | 100.00% |
| Inter-sector differential | 0.2790 | 85.63% | 0.2626 | 91.00% | 0.2681 | 89.10% |
| Explained differential | 0.0720 | 22.11% | 0.0508 | 17.59% | 0.0334 | 11.10% |
| Unexplained differential | 0.2070 | 63.52% | 0.2118 | 73.41% | 0.2347 | 77.99% |
| Intra-sector differential | 0.0468 | 14.37% | 0.0260 | 9.00% | 0.0328 | 10.90% |
| Explained differential | 0.0482 | 14.79% | 0.0542 | 18.77% | 0.0617 | 20.51% |
| Unexplained differential | -0.0014 | -0.42% | -0.0282 | -9.77% | -0.0289 | -9.61% |
| Total explained differentials | 0.1203 | 36.90% | 0.1049 | 36.36% | 0.0951 | 31.62% |
| Total unexplained differentials | 0.2056 | 63.10% | 0.1836 | 63.64% | 0.3298 | 68.38% |

[a] Data were collected from China Urban Household Survey (2004, 2008, 2013)

often being engaged in lower-paying occupations, lacking advanced skills. Wage determination in the private sector is heavily influenced by market forces. Therefore, low-wage individuals usually find employment in labor-intensive industries, while high-wage men predominantly occupy top-tier positions, creating a skewed distribution of gender wage ratios at different quantiles. In the early stages of the collective economy, gender inequality among low-wage individuals was quite significant. This was primarily due to the dominance of labor-intensive agriculture and primary agricultural product processing industries, where men naturally have a physical advantage over women. However, with the modernization of agriculture in China and the popularization of agricultural science and technology [127], the influence of physical gender differences in agricultural production has gradually weakened. This has led to the current gender wage gap in the collective economy remaining stable.

Thirdly, this study shows that the gender wage gap in China mainly stems from inter-sectoral rather than intra-sectoral sources. In other words, the main cause of the gender wage gap is discrimination against women in certain sectors, rather than differences in endowments. This phenomenon may be the result of multiple factors. Firstly, there are significant differences between the typical career choices of males and females in China. Females are more inclined to work in public sector industries such as education and healthcare, while males are more likely to choose fields such as science, technology, engineering and mathematics (STEM) [128]. Secondly, men are more likely to ascend to senior positions in the private sector compared to the collective economy and the public sector. This may be associated with underlying factors such as gender bias, gender discrimination, and an uneven allocation of family responsibilities [129]. Lastly, sector-specific regulations on working conditions and benefits may contribute to gender differences. For example, the public sector tends to offer more substantial benefits and standardized reward mechanisms to women, whereas the private sector pays less attention to female employees, such as in matters of maternity leave [130].

## 6 Conclusion

This study analyzes changes in the gender wage gap within the public sector, private sector, and collective economy in China from 2004 to 2013. It verifies the existence of sectoral segmentation in the Chinese labor market and confirms the continued role of human capital theory. The study concludes that the public sector has consistently exhibited the smallest and most stable gender wage gap of the three sectors. In contrast, the private sector has overtaken the collective economy to become the sector with the largest gender wage gap. The gender wage gap is significant among low-wage groups in all sectors, and a pronounced gender wage gap exists among high-wage individuals within the private sector. Finally, this study finds that differences between sectors, rather than within sectors, are the main cause of the gender wage gap. These differences are mainly attributed to discrimination.

### 6.1 Policy recommendations

The gender wage gap is a complex and systemic social issue, requiring comprehensive and wide-ranging efforts to reduce it.

Firstly, attention must be directed toward the gender wage gap in the private sector. Companies must guarantee that both male and female employees will be compensated equitably for identical roles and establish transparent wage structures and remuneration policies. Concurrently, companies should offer equal opportunities for career training, promotions, and mentorship programs tailored for female employees. Fair promotion criteria must be established to ensure equal opportunities for both genders in their career progression. Moreover, flexible working hours, remote work options, and adaptable work arrangements are viable solutions to

assist female employees in balancing their professional and familial obligations. The government can implement legislation mandating private sector employers to provide fair wages and disclose gender salary data. Simultaneously, there should be stringent oversight on the enforcement of labor laws, penalizing non-compliance in the private sector rigorously.

Secondly, actions should be targeted according to wage groups, with particular focus on the gender wage gap in low-wage occupations. Both the government and society should offer educational and training opportunities aimed at low-wage women, especially in STEM and other high-wage fields, to enhance their employability and earning potential. Companies should facilitate flexible working hours, parental leave, and other support policies to help low-wage women, particularly those in the private sector, to balance work and family responsibilities. In addition, the government could extend social security and welfare benefits like medical insurance, housing subsidies and retirement plans for women. Concurrently, efforts should be made to bolster the enforcement of labor laws to ensure that low-wage women are afforded the same labor rights and protections as men, thereby alleviating their burden.

Thirdly, there is potential to further reduce the gender wage gap in both the public sector and the collective economy. These sectors could serve as exemplary models for recruitment and promotion by establishing fair and unbiased selection criteria and processes, thus ensuring a diverse applicant pool and equal employment opportunities. Additionally, both sectors are well-positioned to develop transparent wage systems, delineate clear pay standards and assessment methods, and conduct regular pay reviews. Furthermore, the public sector and collective economy are better able to gather gender wage data. Through consistent monitoring and evaluation of gender wage gaps, they can develop corrective measures that can be extended to the private sector.

## 6.2 Limitations

Despite utilizing reliable Urban Household Survey (UHS) data from 2004, 2008, and 2013, which includes over 40,000 individual data points, the dataset has two main shortcomings. Firstly, it consists of cross-sectional data rather than panel data, limiting the scope for tracking the evolution of the gender wage gap over time. Secondly, the dataset lacks comprehensive information on wage composition, such as monthly wages and subsidies. This deficiency becomes critical given the varying welfare benefits across sectors in China, resulting in an incomplete picture of the gender wage gap.

Furthermore, the study identifies that the wage gap between males and females is the most substantial among low-wage groups, irrespective of the sector. This discrepancy warrants further research, given its obvious contribution to the overall gender wage gap. Potential areas for future research include the predominance of low-wage workers in labor-intensive industries, as opposed to capital- or knowledge-intensive industries, especially for women with lower educational levels and a lack of awareness of workers' rights.

## Supporting information

**S1 Text. Detailed introduction to the collective economy.**
(DOCX)

**S1 Table. Industry-classification table.**
(DOCX)

**S1 Dataset. The data used in this article.**
(XLS)

## Author Contributions

**Conceptualization:** Mingming Li.

**Data curation:** Mingming Li.

**Formal analysis:** Mingming Li.

**Funding acquisition:** Mingming Li.

**Investigation:** Mingming Li.

**Methodology:** Yuan Tang.

**Project administration:** Keyan Jin.

**Software:** Mingming Li.

**Supervision:** Mingming Li.

**Visualization:** Yuan Tang.

**Writing – original draft:** Mingming Li.

**Writing – review & editing:** Yuan Tang, Keyan Jin.

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
