## [Decision Letter · Decision Letter 0]

23 Oct 2023

PONE-D-23-29332Labor market segmentation and the gender wage gap: evidence from ChinaPLOS ONE

Dear Dr. Li,

Thank you for submitting your manuscript to PLOS ONE. After careful consideration, we feel that it has merit but does not fully meet PLOS ONE’s publication criteria as it currently stands. Therefore, we invite you to submit a revised version of the manuscript that addresses the points raised during the review process.

We look forward to receiving your revised manuscript.

Kind regards,

Keumseok Peter Koh

Academic Editor

PLOS ONE

Journal Requirements:

1. When submitting your revision, we need you to address these additional requirements. Please ensure that your manuscript meets PLOS ONE's style requirements, including those for file naming. The PLOS ONE style templates can be found at https://journals.plos.org/plosone/s/file?id=wjVg/PLOSOne_formatting_sample_main_body.pdf and https://journals.plos.org/plosone/s/file?id=ba62/PLOSOne_formatting_sample_title_authors_affiliations.pdf 2. Did you know that depositing data in a repository is associated with up to a 25% citation advantage (https://doi.org/10.1371/journal.pone.0230416)? If you’ve not already done so, consider depositing your raw data in a repository to ensure your work is read, appreciated and cited by the largest possible audience. You’ll also earn an Accessible Data icon on your published paper if you deposit your data in any participating repository (https://plos.org/open-science/open-data/#accessible-data). 3. Please include captions for your Supporting Information files at the end of your manuscript, and update any in-text citations to match accordingly. Please see our Supporting Information guidelines for more information: http://journals.plos.org/plosone/s/supporting-information.  

Reviewers' comments:

Reviewer's Responses to Questions

**Comments to the Author**

1. Is the manuscript technically sound, and do the data support the conclusions?

Reviewer #1: Yes

2. Has the statistical analysis been performed appropriately and rigorously? 

Reviewer #1: Yes

3. Have the authors made all data underlying the findings in their manuscript fully available?

Reviewer #1: Yes

4. Is the manuscript presented in an intelligible fashion and written in standard English?

Reviewer #1: No

5. Review Comments to the Author

Reviewer #1: The research paper addresses a pressing issue of the gender wage gap in 21st century China, which is both relevant and important for policymakers, economists, and scholars interested in labor market dynamics and gender equality.

The abstract effectively summarizes the key elements of the research, including the problem statement, data sources, methodology, and findings. It provides a concise overview of the study's scope and findings. Utilizing micro-level data from 2004, 2008, and 2013 is a strong point as it allows for a longitudinal analysis of the gender wage gap, offering insights into its trends over time. Employing a selection bias correction based on the multinomial logit model is a rigorous methodological approach and adds credibility to the research. The research appropriately identifies the differences in the gender wage gap within the public sector, private sector, and collective economy. This sector-specific analysis adds depth to the study.

Suggestions for Improvement:

The study should explicitly state the primary research question or objective. This would provide readers with a clearer understanding of the research's focus.

While the study mentions the use of the multinomial logit model, recentered influence function regression, and Brown wage decomposition, consider providing a brief explanation or references for these methodologies. This would be beneficial for readers who may not be familiar with these techniques.

Define the term "collective economy" to ensure clarity for readers who may not be familiar with this concept.

If available, include information on the statistical significance of findings, such as p-values or confidence intervals, to assess the robustness of conclusions.

Ensure that the study is organized logically and follows a clear structure, including the problem statement, data and methodology, key findings, and policy recommendations. This will make it easier for readers to follow the flow of the research.

Please add these relevant references:

https://doi.org/10.3390/sym12020242

https://doi.org/10.1007/s10098-020-02011-w

https://doi.org/10.1007/s10668-019-00365-w

https://doi.org/10.1016/j.spc.2021.11.012

https://doi.org/10.3390/su15097068

https://doi.org/10.1016/j.jclepro.2023.137677

https://doi.org/10.1177/2158244020931899

https://doi.org/10.3390/su151410782

https://doi.org/10.3390/su15118787

Lastly, review the study for grammar, punctuation, and typographical errors to ensure it is well-written and polished.

6. PLOS authors have the option to publish the peer review history of their article (what does this mean?). If published, this will include your full peer review and any attached files.

Reviewer #1: No

---

## [Author Response · Author response to Decision Letter 0]

21 Nov 2023

Dear Editor and Reviewers: 

Thank you for your letter and comments concerning my manuscript entitled “Labor Market Segmentation and the Gender Wage Gap: Evidence from China”. Those comments are extremely valuable and very helpful for revising and improving our paper as well as the significance of our research. 

In terms of format, we have carefully revised the text, tables, figures, and file names according to the requirements of Plos One. In terms of content, we have seriously read the comments and recommended articles, and revised the corresponding parts according to the reviewer opinions. Finally, we used English touch-up services to ensure that the full text was free of grammatical errors and its fluency.

The main responses to comments of editor and the reviewer are as flows. 

To Editor 

The Lines mentioned below are referring to those in Manuscript with no tracks. 

Q1: Please include the following items when submitting your revised manuscript: 'Response to Reviewers', 'Revised Manuscript with Track Changes', 'Manuscript'.

A2: Thank you for your valuable comment. We have finish now renamed all the files with the according to PLOS ONE requirements. 

Q2: Please ensure that your manuscript meets PLOS ONE's style requirements, including those for file naming.

A2: Thank you for your valuable comment. We have carefully read the links to documents as mentioned in your email shown below and adjusted manuscript and figures according to their format requirement. Now all the text, tables and figures match the PLOS ONE's style.

https://journals.plos.org/plosone/s/file?id=wjVg/PLOSOne_formatting_sample_main_body.pdf
https://journals.plos.org/plosone/s/file?id=ba62/PLOSOne_formatting_sample_title_authors_affiliations.pdf

Q3: Please include captions for your Supporting Information files at the end of your manuscript, and update any in-text citations to match accordingly.

A3: Thank you for your valuable comment. At the end of the manuscript, we put the Supporting Information. according to requirement of Plos One (line 818 – line 821): S1 Text, detailed introduction to the Collective Economy; S2 Table, Industry-Classification Table; S3 Dataset, the data used in this article. 

To Reviewer

The Lines mentioned below are referring to those in Manuscript with no tracks. 

Q1: The research paper addresses a pressing issue of the gender wage gap in 21st century China, which is both relevant and important for policymakers, economists, and scholars interested in labor market dynamics and gender equality. The abstract effectively summarizes the key elements of the research, including the problem statement, data sources, methodology, and findings. It provides a concise overview of the study's scope and findings. Utilizing micro-level data from 2004, 2008, and 2013 is a strong point as it allows for a longitudinal analysis of the gender wage gap, offering insights into its trends over time. Employing a selection bias correction based on the multinomial logit model is a rigorous methodological approach and adds credibility to the research. The research appropriately identifies the differences in the gender wage gap within the public sector, private sector, and collective economy. This sector-specific analysis adds depth to the study.

A1: Thank you for your valuable comment. Thank you very much for your positive opinion and giving us this opportunity to make revisions. According to your comments, we have adjusted the introduction section, introduced more on collective economy, given more details about methodologies on measuring inequality, supplement matched important articles. Finally, we double checked the structure using proofreading service to check the grammar, punctuation, and typographical errors to ensure our whole article is well-written and is suitable for publication. Now we believe our paper can make a substantial contribution to literature on gender inequality in China.

Q2: The study should explicitly state the primary research question or objective. This would provide readers with a clearer understanding of the research's focus.

A2: Thank you for your valuable comment. We carefully revised the first two paragraphs (line 34 – line 58). In the first paragraph, we introduce the wage inequality situation shortly, and next, we point out directly that now “the widening gender wage gap has gradually shifted toward sector segmentation theory and related empirical studies”. Further, we briefly introduce why sector segmentation is important in China and suggest systematic research on it currently in China. In last sentence, we directly show our object “this paper tries to understand the role of sector segmentation in the gender wage gap and its change trend in the context of China, addressing the limitations of current research”.

Q3: While the study mentions the use of the multinomial logit model, recentered influence function regression, and Brown wage decomposition, consider providing a brief explanation or references for these methodologies. This would be beneficial for readers who may not be familiar with these techniques.

A3: Thank you for your valuable comment. We carefully adjust empirical methods part to make readers better understand for these methodologies (line 472 – line 573). First, we explain why we use logarithmic form in basic wage function (line 479 – line 481). Then we detail the reasons why it is necessary to use of the multinomial logit model rather using traditional Heckman sample selection model (line 489 – line 496), which is one of our contributions to the literature gap. Third, we further introduce the function of recentered influence function (RIF) model and its regression (line 520 – line 530). Lastly, we introduce and explain how Brown decomposition (line 545 – line 550) can help us to understand the wage gaps by sector segmentation. 

Q4: Define the term "collective economy" to ensure clarity for readers who may not be familiar with this concept.

A4: Thank you for your valuable comment. "Collective Economy" is a very important concept, and we now add much more information about it. In introduction section, we supplement detailed information about the definition of collective economy (line 79 – line 100) to help the reader understand this kind of economy easily under Chinses context. Further, we put more information, especially the characteristics of collective economy in Appendix (supporting information as required by Plos One).

Q5: If available, include information on the statistical significance of findings, such as p-values or confidence intervals, to assess the robustness of conclusions.

A5: Thank you for your valuable comment. We carefully check each table (table 1 -table 5) to make sure that the readers can understand the tables easily. For table 1 – table 2, we double check the data accuracy. For table 3, we put P-value as the bottom line of the table. For table 4 – table 5, as there is not enough place to put P-value and confidence intervals, we make it much clearer in the table note that “Standard error is in parentheses and * p < 0.1, ** p < 0.05, *** p < 0.01” to help the readers to calculate T-value using the regression coefficient/standard error. Based on the T-value, the reader can determine the p-value accordingly easily by themselves. 

Q6: Ensure that the study is organized logically and follows a clear structure, including the problem statement, data and methodology, key findings, and policy recommendations. This will make it easier for readers to follow the flow of the research.

A6: Thank you for your valuable comment. Combing with your suggest and the format requirement of Plos One, now the article is well-structured with following sections: Introduction, Sector Segmentation in China, Literature review, Data and methodology, Empirical Research Results and Discussion, Conclusion. 

Q7: Please add these relevant references:

https://doi.org/10.3390/sym12020242

https://doi.org/10.1007/s10098-020-02011-w

https://doi.org/10.1007/s10668-019-00365-w

https://doi.org/10.1016/j.spc.2021.11.012

https://doi.org/10.3390/su15097068

https://doi.org/10.1016/j.jclepro.2023.137677

https://doi.org/10.1177/2158244020931899

https://doi.org/10.3390/su151410782

https://doi.org/10.3390/su15118787

A7: Thank you for your valuable comment. We have read these important literatures and believe the experience mentioned in theses article can be highly appreciated. Now we cite these articles into our research in different sections. The places of citation can be checked by corresponding reference number. 

Q8: Lastly, review the study for grammar, punctuation, and typographical errors to ensure it is well-written and polished.

A8: Thank you for your valuable comment. We have used the professional proofreading service and double checked the structure to guarantee there is no grammar, punctuation, and typographical errors. Now whole article is fully polished and is suitable for the further publication.

---

## [Decision Letter · Decision Letter 1]

9 Feb 2024

Labor Market Segmentation and the Gender Wage Gap: Evidence from China

PONE-D-23-29332R1

Dear Dr. Li,

We’re pleased to inform you that your manuscript has been judged scientifically suitable for publication and will be formally accepted for publication once it meets all outstanding technical requirements.

Kind regards,

Keumseok Peter Koh

Academic Editor

PLOS ONE

Additional Editor Comments (optional):

Reviewers' comments:

Reviewer's Responses to Questions

**Comments to the Author**

1. If the authors have adequately addressed your comments raised in a previous round of review and you feel that this manuscript is now acceptable for publication, you may indicate that here to bypass the “Comments to the Author” section, enter your conflict of interest statement in the “Confidential to Editor” section, and submit your "Accept" recommendation.

Reviewer #1: All comments have been addressed

2. Is the manuscript technically sound, and do the data support the conclusions?

Reviewer #1: Yes

3. Has the statistical analysis been performed appropriately and rigorously? 

Reviewer #1: Yes

4. Have the authors made all data underlying the findings in their manuscript fully available?

Reviewer #1: Yes

5. Is the manuscript presented in an intelligible fashion and written in standard English?

Reviewer #1: Yes

6. Review Comments to the Author

Reviewer #1: (No Response)

7. PLOS authors have the option to publish the peer review history of their article (what does this mean?). If published, this will include your full peer review and any attached files.

Reviewer #1: No

---

## [Editor Report · Acceptance letter]

18 Mar 2024

PONE-D-23-29332R1 

PLOS ONE

Dear Dr. Li, 

I'm pleased to inform you that your manuscript has been deemed suitable for publication in PLOS ONE. Congratulations! Your manuscript is now being handed over to our production team.

Kind regards, 

on behalf of

Dr. Keumseok Peter Koh 

Academic Editor

PLOS ONE